# Comparison of Current International Guidelines for the Management of Dyslipidemia

**DOI:** 10.3390/jcm11237249

**Published:** 2022-12-06

**Authors:** Sevda Aygun, Lale Tokgozoglu

**Affiliations:** Department of Cardiology, Hacettepe University Faculty of Medicine, 06100 Ankara, Turkey

**Keywords:** dyslipidemia, risk management, secondary prevention, primary prevention

## Abstract

The dyslipidemia guidelines of the three major societies have been revised recently in light of new evidence. LDL-C is the primary target in the ESC, AHA/ACC/Multisociety and Canadian Cardiovascular Society (CCS) guidelines. These guidelines uniformly recommend intensifying lipid-lowering treatment with increased risk; however, the risk estimation systems are different across the guidelines. The ESC guidelines have LDL-C goals which have become more stringent over the years and advocate the use of statin and, if necessary, non-statin therapies to obtain these goals. AHA/ACC/Multisociety guidelines have LDL-C thresholds and advocate combination therapy less liberally and for selected patients. All three guidelines acknowledge the importance of shared decision making. Despite some divergent approaches and recommendations, the main principles and messages are the same across the guidelines. To combat the epidemic of cardiovascular disease, our focus should be not on the differences but on implementing the guidelines in our region.

## 1. Introduction 

Despite modern therapies, atherosclerotic cardiovascular disease (ASCVD) is still the leading cause of mortality in most parts of the world [1]. The retention of apoprotein B-containing lipoproteins is the main driver of the initiation and progression of atherosclerotic plaques [2]. Lowering atherogenic lipids can change the trajectory of the disease favorably and prevent CV events. In light of new evidence on the causality of apoprotein B-containing lipoproteins and, mainly, LDL-C, guidelines for the management of dyslipidemia have been updated. The aim of this article is to systematically compare the 2018 AHA/ACC/Multisociety (MS) Blood cholesterol management guideline [3], the 2021 ESC Prevention of CV Disease Guidelines, endorsed by 12 European Societies [4], and the 2021 Canadian Cardiovascular Society’s (CCS) [5] management of dyslipidemia for cardiovascular disease guidelines for basic approaches to dyslipidemias and the prevention of ASCVD.

## 2. Risk Estimation Tools and Definition of Risk Categories

All three guidelines base the intensity of their recommendations on the degree of risk. However, the best risk estimation system is the one that is derived from the population it is going to be used on. For this reason, guidelines differ in their risk calculation systems. The ESC guidelines define patients with ASCVD, diabetes mellitus (DM), chronic kidney disease (CKD) and individuals with specific risk factors as high and very high-risk groups automatically. Individuals who do not have these characteristics are considered as apparently healthy people, and management is determined according to risk estimation by the SCORE system. The most recent ESC guidelines have updated the risk stratification. SCORE2 is a new algorithm which is derived, calibrated and validated to predict 10-year risk of first-onset CVD in European populations, overcoming some of the limitations posed by the previous SCORE system. The previous SCORE only calculated the 10-year risk of fatal events, whereas SCORE2 calculates the 10-year risk of total CV events. To improve the accuracy of risk prediction in adults over the age of 65, the new SCORE2-Older Persons (SCORE2-OP) model, which is competing-risk-adjusted, is recommended. Management is determined according to age, risk score and region. According to CVD mortality rates published by the WHO, regions are defined into four groups as low-risk countries, moderate-risk countries, high-risk countries, and very high-risk countries (Table 1). 

In the AHA/ACC/MS guidelines, risk scores are calculated with Pooled Cohort Equations (PCEs). PCEs calculate the 10-year risk of developing ASCVD by including non-fatal myocardial infarction or coronary heart disease (CHD) death and fatal or non-fatal stroke, among people free from ASCVD.

The CCS uses the Framingham Risk Score (FRS) system as a risk assessment tool and divides individuals into three groups: low-risk (FRS < 10%), intermediate-risk (FRS 10–19.9%) and high-risk (FRS ≥ 20%), and bases recommendations according to the risk level. The risk stratification methods are summarized in Table 2.

## 3. Risk Modifiers and Risk-Enhancing Factors 

Risk-modifying/enhancing factors are important in making shared decisions regarding treatment initiation and the intensification of recommendations, especially in borderline and low-to-intermediate-risk adults. While there are divergent approaches in the ESC, AHA/ACC/MS and CCS guidelines, the basic approach is to clarify the patient’s current risk level and refine recommendations according to risk modifiers (Table 3). 

Recent studies have demonstrated the importance of the Coronary Artery Calcium (CAC) score to improve risk prediction. The ESC guideline states that CAC scoring may be considered to improve risk classification around treatment decision thresholds. Plaque detection by carotid ultrasound is an alternative when CAC scoring is unavailable or not feasible. AHA/ACC/MS guidelines give recommendations for the possible use of the CAC score if the decision about statin treatment is uncertain in intermediate and borderline-risk adults. If the CAC score is above 100, it is reasonable to initiate statin treatment. If the CAC score is 1 to 99, it is reasonable to use statins in individuals ≥ 55 years. If the CAC score is zero, there is no need to use statins (unless smoking, premature CVD history and DM are present), but reassessment is suggested in 5–10 years.

The CCS suggest that the CAC score might be considered for two major situations: risk classification of asymptomatic adults ≥ 40 years in the intermediate-risk group if the treatment plan is uncertain and for low-risk individuals who have a family history of premature ASCVD events to reevaluate the risk level. It is not recommended to use the CAC score for patients already under statin treatment and asymptomatic low-risk individuals. 

Population studies have shown that some of ethnic groups have a higher risk of CVD events [6]. The ESC guidelines recommend using a country and ethnicity-specific risk calculator. Because of the variation of risk levels between ethnic groups, multiplying the calculated risk level by 1.3 for South Asians, 1.1 for other Asians and by 0.7 for Black African and Chinese populations is recommended. ACC/AHA/MS guidelines underline some racial/ethnic issues and differences between Asian Americans, Hispanic/Latino Americans and Black/African Americans and recommend considering the different ethnic features of individuals to make a decision in treatment and when adjusting the intensity of statin treatment. The CCS recommends earlier screening for some specific groups such as South Asians. It also emphasizes the extremely high level of Lp(a) in the South Asian and Latin American populations. 

## 4. Lipid Measurement

All guidelines recommend LDL-C level measurements as the primary lipid analysis method and recommends using the non-fasting plasma lipid profile for screening in the general population. However, LDL-C levels may be miscalculated in non-fasting measurements in groups who have high triglyceride (TG) levels. For this reason, fasting or the direct measurement of LDL-C is recommended for individuals with high TG levels (especially patients with metabolic syndrome, diabetes mellitus or familial hypertriglyceridemia) [7,8]. AHA/ACC/MS guidelines emphasize fasting lipid profile measurement, especially if TG levels are 400≥mg/dL (or ≥4.5 mmol/L). 

Patients with diabetes mellitus, obesity or metabolic syndrome have a residual lipid risk, which can be captured by non-HDL-C and ApoB measurements [8]. The ESC recommends non-HDL-C and ApoB measurements in all individuals with high TG levels, diabetes mellitus, obesity and metabolic syndrome. Conversely, AHA/ACC/MS guidelines do not routinely recommend ApoB measurement because of cost-effectiveness issues. It emphasizes the importance of ApoB measurement, especially in individuals with TG ≥ 200 mg/dL. The CCS guidelines recommend non-HDL-C or ApoB measurement if LDL-C ≥ 1.5 mmol/L.

The Lp(a) level is a genetically determined, causal and prevalent risk factor for ASCVD. It has been shown that individuals with an Lp(a) level > 180 mg/dL (>430 nmol/L) have a similar ASCVD event risk as individuals with heterozygous FH [9,10]. The ESC guidelines recommend Lp(a) measurements once in each individual’s lifetime. Lp(a) measurement is especially recommended in individuals with a family history of premature ASCVD. Lp(a) levels may also be used to define and reclassify patients in moderate-to-high-risk patients. The AHA/ACC/MS guidelines recommend Lp(a) measurement in individuals with a family history or a history of premature ASCVD and consider a Lp(a) level ≥ 50 mg/dL (125 nmol/L) as a risk-enhancing factor. The CCS guidelines also recommend measuring Lp(a) at least once in a lifetime.

## 5. Primary Prevention

All three guidelines highlight the importance of lifestyle and a heart-healthy diet as the first step in prevention in all individuals. All guidelines also emphasize the importance of being physically active and avoiding a sedentary life. Individuals are encouraged to exercise at a moderate-to-high intensity several times a week. The ESC guidelines also recommend performing resistance exercises 2–3 days a week to reduce all-cause mortality. 

The causality of LDL-C is well established; therefore, it is the primary target of therapy in all guidelines [11]. Since the publication of previous guidelines, large RCTs with combination therapy have shown that lowering LDL-C below 70 mg/dL leads to better CV outcomes in high-risk patients. For this reason, LDL-C goals have become more stringent in the recent ESC guidelines. In addition, Lp(a) measurement is recommended once in a lifetime for all individuals. For a primary prevention in individuals categorized as ‘’the apparently healthy people’’, the ESC guidelines personalize therapy according to the age and SCORE2 risk of the patient and risk modifiers. The ESC guidelines recommend targeting the ultimate goals of ≥50% LDL-C reduction from baseline and an LDL-C goal of <1.4 mmol/L (55 mg/dL) in very high-risk groups, <1.8 mmol/L (<70 mg/dL) in high-risk groups, a goal of <2.6 mmol/L (<100 mg/dL) in moderate-risk groups and a goal of <3.0 mmol/L (<116 mg/dL) in low-risk groups. The guidelines recommend a stepwise approach, with consideration of CVD risk, treatment benefit, comorbidities, frailty and patient preferences. First-line treatment should be a high-intensity statin prescribed up to the highest tolerated dose to reach the LDL-C goals set for the specific risk group. If goals are not achieved, despite maximally tolerated statin dosage, a combination of ezetimibe is recommended. For the very high-risk group, if LDL-C goals are not achieved under statin and ezetimibe treatment, PCSK9 inhibitors are recommended. The ESC guidelines recommend treatment intensification until goals are reached. The ESC guidelines recommend lower LDL-C levels than the recommended treatment thresholds in the AHA/ACC/MS and CCS guidelines. These goals have been determined from the recent trials with combination therapy showing further benefit when LDL-C is lowered beyond 70 mg/dL. 

The AHA/ACC/MS and CCS guidelines recommend starting statin treatment and intensification according to LDL-C thresholds. In the AHA/ACC/MS guidelines, lifestyle changes and healthy behaviors are recommended in low and borderline-risk groups. In the 2018 ACC/AHA/MS guidelines, the statin-benefit groups remain the same as the previous guidelines, but for secondary prevention, the LDL-C threshold has been defined as ≥70 mg/dL, where the addition of a non-statin lipid-lowering drug to statin treatment is recommended. The new guidelines place emphasis on shared decision making and using the calcium score to aid decisions. In the intermediate-risk group, statin initiation is recommended and an LDL-C reduction of 30–49% is targeted. If the patient is in the gray zone for treatment decisions, a Coronary Artery Calcium (CAC) score assessment is reasonable to use for the determination of statin therapy. Moderate-intensity statin therapy is recommended in adults 40 to 75 years of age with diabetes mellitus or LDL-C ≥ 70 to <190 mg/dL. In high-risk groups and in those with an LDL-C level ≥ 190 mg/dL, high-intensity statin initiation is recommended and an LDL-C reduction of ≥50% is targeted. An assessment of response and adherence to treatment after 4–12 weeks and 3–12 months following statin initiation is recommended and, according to the evaluation, treatment intensification is recommended, if needed. Ezetimibe or PCSK9 inhibitors are suggested in a manner of cost effectiveness and shared decision making with patients. In patients at a very high-risk whose LDL-C level remains ≥70 mg/dL (≥1.8 mmol/L) and patients with severe primary hypercholesterolemia on maximally tolerated statin and ezetimibe therapy, adding a PCSK9 inhibitor is deemed reasonable.

The 2022 ACC Expert Consensus Decision Pathway on the Role of Nonstatin Therapies for LDL-Cholesterol Lowering provides additional guidance on the newer non-statin therapies [12]. For adults with ASCVD at very high-risk, if the patient does not have a ≥50% reduction or LDL-C < 55 mg/dL or non-HDL-C<85 mg/dL, despite maximally tolerated statin therapy, ezetimibe and/or PCSK9 inhibitors are recommended as first-line non-statin agents. As the second line of treatment, bempedoic acid and/or inclisiran may be considered. Agents that may be used to treat HoFH under care of a lipid specialist are evinacumab, lomitapide or LDL apheresis.

In primary prevention, if LDL-C is still ≥190 mg/dL, despite maximally tolerated statins, to achieve a ≥50% LDL-C reduction and LDL-C < 100 mg /dL or non-HDL-C < 130 mg/dL, non-statin agents, ezetimibe and/or PCSK-9 inhibitors are the first line of treatment, bempedoic acid or inclisiran are the second line and evinacumab, lomitapide and/or LDL apheresis the third line, respectively.

In adults without ASCVD or diabetes with an LDL-C level of 70–189 mg/dL, and if the patient has a ≥20% risk, and in adults with diabetes without ASCVD and with an LDL-C < 190 mg/dL, if a ≥50%reduction in the LDL-C level or LDL-C < 70 mg/dL or non-HDL-C < 100 mg/dL are not achieved, despite statin therapy, ezetimibe addition may be considered. The conditions requiring treatment intensification with non-statin agents are shown in Table 4.

Screening recommendations have continued in the CCS guidelines, recommending blood cholesterol screening in all individuals aged ≥40 years or with risk factors. Thresholds have been defined for treatment initiation and intensification. CAC score measurement is recommended for screening in asymptomatic and intermediate-risk patients ≥40-years-old. In the CCS guidelines, for primary prevention, patients are divided into three groups according to the FRS. In low-risk groups, lifestyle changes are first-line recommendations, and statin initiation is not recommended. Individuals without high-risk conditions who may benefit from statin therapy are the following: (a) LDL-C ≥ 5.0 mmol/L or Apo B ≥ 1.45 g/L or non-HDL-C ≥ 5.8 mmol/L, (b) FRS 5–9.9% with LDL-C ≥ 3.5 mmol/L or non-HDL-C ≥ 4.2 mmol/L or ApoB ≥ 1.05 g/L, with additional risk modifiers such as Lp(a) ≥ 50 mg/dL, CAC > 0 AU and familial or genetic dyslipidemias. Statin treatment is recommended along with lifestyle changes for intermediate-risk individuals (FRS 10–19%) with LDL-C ≥ 3.5 mmol/L and high-risk patients (FRS ≥ 20%). Despite maximally tolerated statin dose, if LDL-C ≥ 2.0 mmol/L, or ApoB ≥ 0.8 g/L or non-HDL-C > 2.6 mmol/L are present, treatment intensification is recommended with ezetimibe. 

All guidelines agree that therapy needs to be intensified in patients as the risk increases. The major difference between these guidelines is that there are defined LDL-C goals in the European guidelines, which are more stringent for patients at high risk or above compared to other guidelines. All guidelines agree that statins are recommended as the first-line treatment, and non-statin treatment (ezetimibe and PCSK-9 inhibitors) are the second-line treatment. The AHA/ACC/MS and CCS guidelines recommend cost-effective approaches for treatment intensification in primary prevention. 

## 6. Secondary Prevention 

All guidelines recommend immediate lipid-lowering treatment initiation in secondary prevention. The ESC guidelines define patients who have ASCVD to be automatically at very high-risk and recommends at least a 50% reduction from baseline, with a goal of below 55 mg/dL. If the patient experiences a recurrent ASCVD event within 2 years after the first event, an LDL-C goal below 40 mg/dL may be considered. After high-intensity statin initiation, patients are evaluated in 4–6 weeks for treatment response. If the LDL-C level is above 55 mg/dL, despite maximally tolerated statin dosage, the addition of ezetimibe or initiation of PCSK9 inhibitors after ezetimibe is recommended for add-on therapy. The ESC guidelines are more liberal in recommending non-statin therapies to obtain the goal. In addition to lipid lowering, the ESC guidelines have introduced the consideration of anti-inflammatory therapy in the form of low-dose colchicine (0.5 mg o.d.) in patients with ASCVD, poorly controlled risk factors or those who experience recurrent events on optimal medical therapy, according to new studies [13]. 

The AHA/ACC/MS guidelines recommend high-intensity statin treatment and a 50% reduction in LDL-C level or, if not tolerated, moderate-intensity statin treatment and a 30–49% reduction in the LDL-C level in high-risk patients with ASCVD. If the desired reduction is not achieved, the first option is the addition of ezetimibe. If LDL-C levels are 70 mg/dL (1.8 mmol/L) or higher or the non–HDL-C level is 100 mg/dL (2.6 mmol/L) or higher under the statin and ezetimibe combination, the addition of an PCSK9 inhibitor may be considered.

The CCS guidelines also recommend high-intensity statin treatment for secondary prevention. If LDL-C remains ≥1.8–2.2 mmol/L or non-HDL-C ≥ 2.4–2.9 mmol/L or ApoB ≥ 0.7–0.8 g/L, while receiving the maximally tolerated statin dose, PCSK9 inhibitors with or without ezetimibe are recommended. If LDL-C remains ≥ 2.2 mmol/L or non-HDL-C ≥ 2.9 mmol/L or ApoB ≥ 0.8 g/L, while receiving the maximally tolerated statin dose, PCSK9 inhibitors with or without ezetimibe are recommended.

## 7. Very High-Risk Patients 

There is no universal consensus on the definition of very high-risk patients, but it is recommended to intensify preventive approaches for these patients in all guidelines. The very high-risk patient category definition is different between guidelines (Table 5).

## 8. Familial Hypercholesterolemia

LDL-C is not only causal but also has a cumulative effect. There is a logarithmic increase between the exposure time and the risk of developing ASCVD. Earlier intervention prevents LDL-C accumulation and changes the trajectory of the disease. Patients with familial hypercholesterolemia (FH) have genetically elevated LDL-C levels and are exposed to elevated LDL-C from early on in life [14]. It is particularly important to diagnose FH early and start treatment. The ESC guidelines automatically classify individuals with FH as being at high-risk and recommend a ≥50% reduction from baseline, with an LDL-C goal of <70 mg/dL. If individuals have FH and one or more additional risk factor such as diabetes mellitus, coronary artery disease or chronic kidney disease, they are classified as being at very high-risk, and the goal is a ≥50% reduction from baseline and an LDL-C goal of <1.4 mmol/L (55 mg/dL). To reach this goal, maximally tolerated statin treatment and, if not at goal, a combination with ezetimibe, is recommended. PCSK-9 inhibitors may be added into therapy if the goal is still not reached. The AHA/ACC/MS guidelines define patients with primary severe hypercholesterolemia (LDL-C levels ≥ 190 mg/dL [≥4.9 mmol/L]) as a statin-benefit group with a high-risk of ASCVD, recommending high-intensity statins. If the LDL-C level is above 2.6 mmol/L (>100 mg/dL), despite statins, it is deemed reasonable to add ezetimibe. If LDL-C is still above 100 mg/dL, the addition of PCSK-9 inhibitors may be considered. The CCS guidelines categorize FH patients as being at high-risk and having a condition requiring statins. If the LDL-C level is above 2.5 mmol/L, despite statins, ezetimibe or PCSK9 inhibitors may be added. PCSK-9 inhibitors are recommended in the following patients: (a) In heterozygous FH patients without clinical ASCVD and LDL-C levels ≥2.5 mmol/L, if a ≥50% reduction of LDL-C levels, or ApoB ≥ 0.85 g/L or non-HDL-C ≥ 3.2 mmol/L. (b) In heterozygous FH patients with ASCVD whose target LDL-C levels remain ≥1.8 mmol/L, or ApoB ≥ 0.7 g/L or non-HDL-C ≥ 2.4 mmol/L, despite a maximally tolerated statin and ezetimibe combination.

## 9. Other Specific Groups

### 9.1. Diabetes Mellitus 

In all the guidelines, diabetes is given special consideration. The ESC guidelines divide diabetic patients into three categories according to concomitant risk factors, target organ damage and age. Patients with well-controlled short-duration diabetes (no evidence of target organ damage or ASCVD risk factors) are classified as moderate-risk; patients without ASCVD or target organ damage not fulfilling moderate-risk criteria are high-risk; while patients with at least three risk factors or type 1 diabetes of a >20 years duration are classified as very-high risk. Other patients between very high and moderate-risk groups are identified as high-risk groups. LDL-C goals depend on the risk. The AHA/ACC/MS guidelines divide diabetes patients into moderate or high-risk groups and recommend moderate-intensity statin treatment to all patients with diabetes. In diabetics at a higher risk, especially those with multiple risk factors or those 50 to 75 years of age, it is deemed reasonable to use a high-intensity statin to reduce the LDL-C level by ≥50%. The CCS considers patients ≥ 40 years of age and patients ≥ 30 years with a 15-year or more duration of diabetes or with microvascular complications to be at high-risk and recommends statin initiation initially, with add-on ezetimibe if necessary.

### 9.2. Chronic Kidney Disease

The ESC guidelines define CKD patients to be at high risk (eGFR 30–59 mL/min per 1.73 m^2^) and very high-risk (eGFR < 30 mL/min per 1.73 m^2^). Statins or statin–ezetimibe use is recommended in all CKD patients not on dialysis. The AHA/ACC/MS guidelines define CKD (estimated glomerular filtration rate 15–59 mL/min per 1.73 m^2^) as a risk enhancing factor and underline that statin initiation is reasonable in patients not treated with dialysis or renal transplantation. The CCS recommends statin initiation to all patients with a GFR < 60 mL/min/1.73 m^2^ and with a preserved GFR but who have an increased urinary albumin-to-creatinine ratio (≥3 mg/mmol) for at least 3 months. The CCS guidelines define patients with CKD (>50 years) as being in the in high-risk category and recommends statin and/or ezetimibe therapy for patients not treated with dialysis or who have a kidney transplantation (patients with eGFR < 60 mL/min/1.73 m^2^ and preserved eGFR). In all guidelines, statin continuation is recommended in patients treated with hemodialysis who are already on statins, but statin initiation is not recommended. 

### 9.3. Hypertriglyceridemia

In the ESC guidelines, there are no TG goals, but TG level <1.7 mmol/L (<150 mg/dL) indicates a lower cardiovascular risk. To address atherogenic triglyceride-rich lipoproteins such as remnants, the ESC guidelines have secondary goals of non-HDL-C < 2.2, 2.6, and 3.4 mmol/L (<85, 100, and 130 mg/dL) for very-high, high, and moderate-risk people, respectively. ApoB secondary goals are <65, 80, and 100 mg/dL for very high, high, and moderate-risk people, respectively.

The ESC guidelines recommend statin treatment as the first line of treatment in high-risk individuals with plasma fasting TG levels > 2.3 mmol/L (>200 mg/dL), despite lifestyle changes. In high-risk patient groups who have achieved LDL-C goals but have TG levels > 2.3 mmol/L (>200 mg/dL), fibrates may be considered in addition to statin treatment. Furthermore, the ESC guidelines recommend considering the combination of n-3 PUFAs (icosapent ethyl 2 g twice a day) with statins in high and very high-risk patient groups with TG levels between 1.5 and 5.6 mmol/L (135–499 mg/dL).

The AHA/ACC/MS guidelines recommend optimizing diet and lifestyle as the first step, ruling out secondary causes of hypertriglyceridemia, and considering statin therapy in those with moderate hypertriglyceridemia and elevated 10-year ASCVD risk. The more recent ACC 2021 expert consensus on the management of ASCVD risk reduction in patients with persistent hypertriglyceridemia also recommends considering icosapent ethyl in high-risk patients [15].

The CCS also recommends the use of high-dose icosapent ethyl to decrease the risk of CV events in patients with ASCVD, or with diabetes and ≥1 CVD risk factors, who have an elevated fasting triglyceride level of 1.5–5.6 mmol/L, despite treatment with maximally tolerated statin therapy.

## 10. Conclusions

The ESC, AHA/ACC/MS and CCS guidelines are based on the principle that LDL-C lowering is a key strategy to prevent CV events. Although divergent interpretations of the evidence result in some differences in treatment recommendations, the main principles are similar [16]. All guidelines strongly advocate that LDL-C should be our primary target and the intensity of treatment should increase as the risk of the patient increases. The ESC guidelines take into account contemporary evidence from combination therapy and imaging trials, setting more stringent LDL-C goals for high-risk patients than any other guidelines. The validity and safety of this approach have been demonstrated by the recent FOURIER-OLE trial [17]. Furthermore, having LDL-C goals motivates the patient and the physician. The shared decision-making approach, as well as using imaging for risk discrimination, recommended in the AHA/ACC/MS and CCS guidelines, is an important step forward. Instead of focusing on differences, we should aim to implement guidelines as much as possible. A universal problem is the under implementation of the guidelines and nonadherence to lifestyle and medications. Real-life registries all over the world highlight the underuse of statins in high doses and combination therapy, as well as discontinuation of medications, resulting in the underachievement of goals. Euroaspire III, IV and V studies have provided important information about the under implementation of guidelines and the underachievement of goals across Europe [18,19]. Only a third of the patients achieved their LDL-C goals in Euroaspire V [20]. The more recent Da Vinci trial confirmed these findings and also pointed out the underutilization of combination therapy and high-intensity statins [21].

We are entering a new era of precision medicine, with the aim of delivering the right treatments, at the right time, to the right person [22]. Lifelong exposure to CVD risk factors is better captured by genetic susceptibility since genetic risk is accumulated continuously over a person’s life span [23,24]. The future of risk prediction and management lies in shifting from population-based risk scores towards personalized risk prediction, where genetic, omics and imaging information is integrated to personalized lifetime risk prediction and management.

The significant reductions in cardiovascular events that we see in trials can be achieved in real-world patient care if we are able to significantly improve the implementation of the evidence-based treatments and achieve recommended lipid targets based on these and other international guidelines. 

## Figures and Tables

**Table 1 jcm-11-07249-t001:** Classification of countries according to risk levels described by the WHO.

Risk Categories	Countries
Low-risk	Belgium, Denmark, France, Israel, Luxembourg, Norway, Spain, the Netherlands, the United Kingdom, Switzerland
Moderate-risk	Austria, Cyprus, Finland, Germany, Greece, Iceland, Ireland, Italy, Malta, Portugal, San Marino, Slovenia, and Sweden
High-risk	Albania, Bosnia and Herzegovina, Croatia, Czech Republic, Estonia, Hungary, Kazakhstan, Poland, Slovakia, and Turkey
Very high-risk	Algeria, Armenia, Azerbaijan, Belarus, Bulgaria, Egypt, Georgia, Kyrgyzstan, Latvia, Lebanon, Libya, Lithuania, Montenegro, Morocco, Republic of Moldova, Romania, Russian Federation, Serbia, Syria, The Former Yugoslav Republic (Macedonia), Tunisia, Ukraine, and Uzbekistan

**Table 2 jcm-11-07249-t002:** Comparison of risk categories according to guidelines.

	ESC GUIDELINES	AHA/ACC/MS GUIDELINES	CCS GUIDELINES
RISK CATEGORIES	10-year SCORE2/SCORE2-OP percentages (fatal and non-fatal CVD risk) **<50 years:** <2.5%, 2.5–7.5%, ≥7.5% **50–69 years:** <5%, 5–10%, ≥10% **≥70 years:** <7.5%, 7.5–15%, ≥15% (Low-to-moderate-risk, high-risk and very high-risk, respectively)	10-year risk ASCVD percentages (fatal and non-fatal ASCVD) **High:** ≥20% **Intermediate:** ≥7.5–<20% **Borderline:** 5–<7.5% **Low:** <5%	FRS 10-year CHD RİSK **Low-risk** FRS: <10% **Intermediate-risk** FRS: 10–19.9% or LDL-C ≥ 3.5 mmol/L or Non-HDL-C ≥ 4.2 mmol/L or ApoB ≥ 1.05 g/L or Men ≥ 50 and women ≥ 60 years with additional risk factors or with presence of other risk modifiers **High-risk** FRS: ≥ 20%

**Table 3 jcm-11-07249-t003:** Risk-modifying and enhancing factors.

ESC Risk Modifiers	AHA/ACC/MS Risk-Enhancing Factors	CCS Risk Modifiers
Family history of premature CVD (men: <55 years and women: <60 years)	Family history of premature ASCVD (males: <55 years; females: <65 years)	Family history of premature coronary artery disease
Obesity and central obesity	ABI < 0.9	Abdominal obesity
Physical inactivity Social deprivation and psychosocial stress, including vital exhaustion.	High-risk race/ethnicities (e.g., South Asian ancestry)	Physical inactivity Psychosocial factors
Chronic immune-mediated inflammatory disorder. Treatment for human immunodeficiency virus infection. Major psychiatric disorders Left ventricular hypertrophy. Chronic kidney disease. Atrial fibrillation Obstructive sleep apnea syndrome Non-alcoholic fatty liver disease Migraine with aura	Metabolic syndrome Primary hypercholesterolemia (LDL-C, 160–189 mg/dL [4.1–4.8 mmol/L); non–HDL-C 190–219 mg/dL [4.9–5.6 mmol/L]) Persistently elevated, primary hypertriglyceridemia (≥175 mg/dL) optimally, three determinations Chronic kidney disease Chronic inflammatory conditions such as psoriasis, RA, or HIV/AIDS	Excessive alcohol consumption
		Coronary Artery Calcium score [CAC] > 0 Agatston Units (AUs)
Sex-specific conditions: Pregnancy-related hypertension Preeclampsia/Eclampsia Erectile dysfunction	Biomarkers Elevated high-sensitivity C-reactive protein (≥2.0 mg/L) Elevated Lp(a): A relative indication for its measurement is family history of premature ASCVD. An Lp(a) ≥ 50 mg/dL or ≥125 nmol/L constitutes a risk-enhancing factor, especially at higher levels of Lp(a) Elevated ApoB ≥ 130 mg/dL: A relative indication for its measurement would be triglyceride ≥200 mg/dL. A level ≥ 130 mg/dL corresponds to an LDL-C ≥ 160 mg/dL and constitutes a risk-enhancing factor.	Biomarkers High-sensitivity C-reactive protein ≥ 2.0 Mmol/L High Lipoprotein(A) [Lp(a)] ≥ 50 mg/dL [≥100 Nmol/L]
	Sex-specific Conditions: Premature menopause (before age of 40) Pregnancy-associated conditions (preeclampsia, eclampsia)	Sex-pecific conditions: Pregnancy-related hypertension Preeclampsia/eclampsia

**Table 4 jcm-11-07249-t004:** LDL-C goals and thresholds for beginning combination therapy with non-statin agents.

	PRIMARY PREVENTION	SECONDARY PREVENTION
**ESC** **Guidelines**	Despite maximally tolerated statin dosage, ≥50% LDL-C reduction from baseline and LDL-C goal of <1.4 mmol/L (55 mg/dL) in very high-risk groups, <1.8 mmol/L (<70 mg/dL) in high-risk groups, <2.6 mmol/L (<100 mg/dL) in moderate-risk groups <3.0 mmol/L (<116 mg/dL) in low-risk groups is not achieved, treatment intensification with non-statin agents is recommended.	If LDL-C ≥ 55 mg/dL, despite maximally tolerated statin dosage, addition of ezetimibe or PCSK9 inhibitors after ezetimibe initiation is recommended.
**AHA/ACC/MS Guideline ***	In adults without ASCVD or diabetes with LDL-C level of 70–189 mg/dL, if patient has ≥20% risk, and In adults with diabetes without ASCVD and with LDL-C < 190 mg/dL, if ≥50% reduction in LDL-C level or LDL-C < 70 mg/dL or non-HDL-C < 100 mg/dL are not achieved, despite statin therapy, ezetimibe additon may be reasonable. In adults without ASCVD and LDL-C ≥ 190 mg/dL, if ≥50% reduction in LDL-C level or LDL-C < 100 mg/dL or non-HDL-C < 130 mg/dL are not achieved, despite statin therapy, non-statin agents are recommended.	Patients with ASCVD and at very high-risk adults with ASCVD at very high-risk, if ≥50% reduction of LDL-C level or LDL-C < 55 mg/dL are not achieved despite statin therapy, non-statin agents are recommended. For patients with ASCVD but without very high- risk, if ≥50% reduction of LDL-C level or LDL-C < 70 mg/dL are not achieved despite statin therapy non-statin agents are recommended.
**CCS Guideline**	Despite maximally tolerated statin dose, LDL-C ≥ 2.0 mmol/L or ApoB ≥ 0.8 g/L or Non-HDL-C ≥ 2.6 mmol/L, ezetimibe and/or PCSK-9 inhibitors are recommended; Despite maximally tolerated statin dose with or without ezetimibe, for patients with heterozygous FH without clinical ASCVD, if LDL-C ≥ 2.5 mmol/L or <50% reduction from baseline; or ApoB ≥ 0.85 g/L or non-HDL-C ≥ 3.2 mmol/L) PCSK-9 inhibitors are recommended.	Despite maximally tolerated statin dose, LDL-C ≥ 1.8–2.2 mmol/L or ApoB ≥ 0.7–0.8 g/dL or Non-HDL-C ≥ 2.4–2.9 mmol/L PCSK9 inhibitors with or without ezetimibe ar recommended. Despite maximally tolerated statin dose, LDL-C ≥ 2.2 mmol/L or ApoB ≥ 0.8 g/L or Non-HDL-C ≥ 2.9 mmol/L, PCSK9 inhibitors with or without ezetimibe are recommended.

* Based on the 2022 ACC Expert Consensus Decision Pathway on the Role of Nonstatin Therapies for LDL-Cholesterol Lowering in the Management of Atherosclerotic Cardiovascular Disease Risk.

**Table 5 jcm-11-07249-t005:** Definitions of very high-risk patients.

ESC GUIDELINES	AHA/ACC/MS GUIDELINES	CCS GUIDELINES
To have one of these conditions below	Two or more major ASCVD events OR One major event and >1 high-risk condition	To have one of these conditions below
• Documented clinical ASCVD • Unequivocal ASCVD on imaging predictive of ASCVD events • Type 2 diabetes mellitus with target organ damage (microalbuminuria, retinopathy, or neuropathy), or at least three major risk factors, or early onset T1DM of long duration (>20 y) • Severe CKD (eGFR < 30 mL/min per 1.73 m^2^). • A calculated SCORE ≥ 10% or 10-year risk of fatal CVD • FH with ASCVD or with another major risk factor	Major ASCVD events	Recent acute coronary event (ACS): • Hospitalized index ACS to 52 weeks post index ACS Clinically evident ASCVD and any of the following: • Diabetes mellitus or metabolic syndrome • Polyvascular disease (vascular disease in ≥2 arterial beds) • Symptomatic PAD • Recurrent MI • MI in the past 2 years • Previous CABG surgery • LDL-C ≥ 2.6 mmol/L or heterozygous FH • Lipoprotein(a) ≥ 60 mg/dL (120 nmol/L) High-risk conditions for primary prevention: • CKD • Diabetes mellitus in patients > 40 years or patients > 30 years and with 15 or more years’ duration of diabetes or with microvascular complications • Abdominal aortic aneurysm > 3.0 cm or previous aortic aneurysm surgery.
	• Recent ACS (within the past 12 months) • History of MI (other than the recent ACS event listed above) • History of ischemic stroke • Symptomatic peripheral arterial disease (history of claudication with ABI <0.85, or previous revascularization or amputation)	
	High-risk conditions	
	• Age ≥ 65 years • Diabetes mellitus • Hypertension • CKD (eGFR 15–59 mL/min per 1.73 m^2^) • History of congestive heart failure • Current smoking • Heterozygous FH • History of prior coronary artery bypass surgery or percutaneous coronary intervention outside of the major ASCVD event(s) • Persistently elevated LDL-C ≥ 100 mg/dL (2.6 mmol/L), despite maximally tolerated statin therapy and ezetimibe	

## Data Availability

Not applicable.

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
