# Peer review of "Comparison of Current International Guidelines for the Management of Dyslipidemia"

_jcm, 2022, doi:10.3390/jcm11237249_

Round 1
Reviewer 1 Report
The paper systematically compare the current International Guidelines for the Management of Dyslipidemia, espeically the ESC, AHA/ACC/ Multisociety and Canadian Cardiovascular Society (CCS) guidelines. The author stated that the aim of the paper was not on the differences but on implementing the guidelines in their region.
1. In the paper, the author only compared the difference of guidelines from different counties. Whether these latest guidelines have made any content modification compared with the old version guidelines are not described in this paper. In fact, through the description of these modifications and revisions of guidelines, we can know whether the content of the latest guidelines has been updated. And these modifications exactly reflect the latest research progress in this field.
2. Since the author stated that the focus of this paper was not on the differences but on implementing the guidelines in their region. It is necessary to state how these guidelines were implemented in their country, and what the difference of their own guidelines with these international guidelines.
3. I think the aim to study the guidelines from different countries was to help us to update, modify or perfect our own guideline. So, it is important for authors to tell in the paper what enlightenments have the author obtained from comparing these guidelines.
Author Response
FIRST REPORT
Thank you for your review and relevant questions. You can see our answers below:
STATEMENT 1: In the paper, the author only compared the difference of guidelines from different counties. Whether these latest guidelines have made any content modification compared with the old version guidelines are not described in this paper. In fact, through the description of these modifications and revisions of guidelines, we can know whether the content of the latest guidelines has been updated. And these modifications exactly reflect the latest research progress in this field.
ANSWER: Thank you for your evaluation. The purpose of our article was to compare the most recent version of the three guidelines therefore we did not go into detail about what was the change compared to previous guidelines to keep within the word count. We have now added the following statements about the most relevant changes:
The most recent ESC guideline has updated the risk stratification. SCORE2 is a new algorithm which is derived, calibrated, and validated to predict 10-year risk of first-onset CVD in European populations overcoming some of the limitations posed by the previous SCORE system. The previous SCORE only calculated the 10-year risk of fatal events whereas SCORE2 calculates 10-year risk of total CV events. (Page 1, paragraph 3)
Since the publication of previous guidelines, large randomized trials on combination therapy have shown that lowering LDL-C below 70 mg/dL leads to better CV outcomes in high risk patients. For this reason, LDL-C goals have become more stringent in the recent ESC guidelines. In addition, Lp(a) measurement is recommended once in a lifetime for all individuals. (Page 4, paragraph 5)
In the 2018 ACC/AHA/MS guidelines the statin benefit groups remain the same as the previous guideline but for secondary prevention, LDL-C threshold has been defined as ≥70 mg/dL where addition of a non-statin lipid-lowering drug to statin treatment is recommended. The new guideline places emphasis on shared decision making and using calcium score to aid decision. (Page 4, Paragraph 6)
Screening recommendations have changed in the CCS guidelines recommending blood cholesterol screening in all individuals aged ≥40 years or with risk factors. Thresholds have been defined for treatment initiation and intensification. CAC score measurement is recommended for screening in asymptomatic and intermediate risk patients ≥40 years. (Page 5, Paragraph 2)
STATEMENT 2: Since the author stated that the focus of this paper was not on the differences but on implementing the guidelines in their region. It is necessary to state how these guidelines were implemented in their country, and what the difference of their own guidelines with these international guidelines.
ANSWER: We endorse the European Society of Cardiology guidelines for dyslipidemia and ASCVD risk management in our country and do not advocate any other guideline. In terms of guideline implementation, we have the same limitations like the rest of Europe as demonstrated by the EUROASPIRE III, IV and V studies performed in 28 European countries that we also participated in. This has been added to the text as follows:
Real life registries all over the world highlight the underuse of statins in high doses and combination therapy as well as discontinuation of medications, resulting in underachievement of goals. Euroaspire III, IV and V studies have provided important information about under implementation of guidelines and underachievement of goals across Europe. Only a third of the patients achieved their LDL-C goals in Euroaspire V. The more recent Da Vinci trial confirmed these findings and also pointed out the underutilization of combination therapy and high intensity statins. (Page 8, Paragraph 3)
STATEMENT 3: I think the aim to study the guidelines from different countries was to help us to update, modify or perfect our own guideline. So, it is important for authors to tell in the paper what enlightenments have the author obtained from comparing these guidelines.
ANSWER: Thank you for this input. We have added the following sentence to the paper to highlight the most important learnings:
All guidelines strongly advocate that LDL-C should be our primary target and intensity of treatment should increase as the risk of the patient increases. Recent trials have shown that the lower LDL-C levels decrease CV risk without safety concerns. The ESC guideline is taking into account contemporary evidence from combination therapy and imaging trials and have set more stringent LDL-C goals for high-risk patients than any other guideline. The validity and safety of this approach has been demonstrated by the recent FOURIER-OLE trial. Furthermore, we believe that having LDL-C goals will motivate the patient and the physician. The shared decision-making approach as well as using imaging for risk discrimination recommended in the ACC/AHA/MS and CCS guideline is an important step forward. (Page 8, Paragraph 3)

Reviewer 2 Report
In this review article, the authors nicely summarized the current guidelines from Europe, the US, and Canada. The essential points look similar, although they used different types of risk calculators. I think this is well written. However, this reviewer raises a couple of issues need to be discussed.
“Guideline” can only accept the data from so-called “evidence”. However, recent studies have suggested that personalized medicine based on human genome will be quite useful in the preventive cardiology as well. As far as I understand, AHA/ACC recently published a statement regarding this issue. It would be nice to refer to this future direction.
The management for familial hypercholesterolemia (FH) should be stipulated, since the management for this special, but common situation is important and different. In particular, the earlier diagnosis looks like very good in this situation. I wonder how this can be true in general population (is earlier, the better?).
The authors compared 3 guidelines. I wonder what are the main differences between them, and which one is the best?
Author Response
SECOND REPORT
Thank you for your review and questions. You can see our answers below.
STATEMENT 1: “Guideline” can only accept the data from so-called “evidence”. However, recent studies have suggested that personalized medicine based on human genome will be quite useful in the preventive cardiology as well. As far as I understand, AHA/ACC recently published a statement regarding this issue. It would be nice to refer to this future direction.
ANSWER:
The following has been added to the text:
We are entering a new era of precision medicine with the aim of delivering the right treatments, at the right time to the right person. Lifelong exposure to CVD risk factors is better captured by genetic susceptibility since genetic risk is accumulated continuously over a person’s life span. The future of risk prediction lies in shifting from population-based risk scores towards personalized risk prediction where genetic, omics and imaging information is integrated to personalized lifetime risk prediction and management. (Page 8, Paragraph 4)
STATEMENT 2: The management for familial hypercholesterolemia (FH) should be stipulated, since the management for this special, but common situation is important and different. In particular, the earlier diagnosis looks like very good in this situation. I wonder how this can be true in general population (is earlier, the better?).
ANSWER: The point is well taken, and we have added a new section section on FH:
LDL-C is not only causal but also has a cumulative effect. There is a logarithmic increase between the exposure time and the risk of developing ASCVD. Earlier intervention will prevent LDL-C accumulation and change the trajectory of the disease. Patients with familial hypercholesterolemia (FH) have genetically elevated LDL-C levels and are exposed to elevated LDL-C from early on in life. It is particularly important to diagnose FH early and start treatment.
ESC guidelines automatically classify individuals with FH as having high risk and recommend a ≥50% reduction from baseline with an LDL-C goal of <70 mg/dL. If individuals have FH and one or more additional risk factor such as diabetes mellitus, coronary artery disease or chronic kidney disease they are classified as having very high-risk and the goal is ≥50% reduction from baseline and LDL-C goals of <1.4 mmol/L (55mg/dL). To get to goal, maximally tolerated statin treatment and if not at goal; combination with ezetimibe is recommended. PCSK-9 inhibitors may be added into therapy if still not at goal.
The AHA/ACC/MS guideline defines patients with primary severe hypercholesterolemia (LDL-C levels ≥190 mg/dL [≥4.9 mmol/L]) as a statin benefit group with a high-risk of ASCVD recommending high intensity statins. If LDL-c level is above 2.6 mmol/L (>100 mg/dL) despite statins, it is deemed reasonable to add ezetimibe. If LDL-C is still above 100 mg/dL, addition of PCSK-9 inhibitors may be considered.
CCS guidelines categorize FH patients as having high risk and a statin indication condition. If LDL-C level is above 2.5 mmol/L despite statins, ezetimibe or PCSK9 inhibitors may be added. PSCK-9 inhibitors are recommended in the following patients: a) In heterozygous FH patients without clinical ASCVD and LDL-c levels, if ≥50% reduction of LDL-c levels, or ApoB ≥ 0.85 mg/dL or non-HDL-c ≥ 3.2 mmol/L. b) In heterozygous FH patients with ASCVD whose target LDL-c levels remain ≥1.8 mmol/L or ApoB ≥0.7 mg/dL or non-HDL- ≥2.4 mmol/L despite maximally tolerated statin and ezetimibe combination. (Page 6, Paragraph 3)
STATEMENT 3: The authors compared 3 guidelines. I wonder what are the main differences between them, and which one is the best?
ANSWER: The risk stratification systems are different between guidelines. As mentioned in the text, the best risk estimation system is the one that is derived from the population it is going to be used on. The ESC have set specific goals for LDL-C. We believe that having goals motivates the patients and increases adherence to treatment. The LDL-C goals are more stringent in the European guidelines reflecting contemporary evidence and are more aligned with the biology of disease.
All guidelines strongly advocate that LDL-C should be our primary target and intensity of treatment should increase as the risk of the patient increases. Recent trials have shown that the lower LDL-C levels decrease CV risk without safety concerns. The ESC guideline is taking into account contemporary evidence from combination therapy and imaging trials and have set more stringent LDL-C goals for high-risk patients than any other guideline. The validity and safety of this approach has been demonstrated by the recent FOURIER-OLE trial. Furthermore, we believe that having LDL-C goals will motivate the patient and the physician. The shared decision-making approach as well as using imaging for risk discrimination recommended in the ACC/AHA/MS and CCS guideline is an important step forward. (Page 8, Paragraph 3)

Round 2
Reviewer 2 Report
I have no additional comment.
Author Response
thank you